# Delayed Ischemic Neurological Deficit after Uneventful Elective Clipping of Unruptured Intracranial Aneurysms

**DOI:** 10.3390/brainsci10080495

**Published:** 2020-07-29

**Authors:** Petr Vachata, Jan Lodin, Aleš Hejčl, Filip Cihlář, Martin Sameš

**Affiliations:** 1Department of Neurosurgery, J. E. Purkyně University, Masaryk Hospital, EU 401 13 Ústí nad Labem, Czech Republic; jan.lodin@kzcr.eu (J.L.); ales.hejcl@kzcr.eu (A.H.); martin.sames@kzcr.eu (M.S.); 2Department of Neurosurgery, University Hospital in Pilsen, The Faculty of Medicine in Pilsen, Charles University in Prague, 30605 Prague, Czech Republic; 3Department of Radiology, J. E. Purkyně University, Masaryk Hospital, EU 401 13 Ústí nad Labem, Czech Republic; filip.cihlar@kzcr.eu

**Keywords:** unruptured intracranial aneurysm, vasospasm, delayed ischemic neurological deficit

## Abstract

Cerebral vasospasm and subsequent delayed ischemic neurological deficit is a typical sequela of acute subarachnoid hemorrhage after aneurysm rupture. The occurrence of vasospasms after uncomplicated surgery of an unruptured aneurysm without history of suspected rupture is extremely rare. The pathogenesis and severity of cerebral vasospasms is typically correlated with the amount of blood breakdown products extravasated during subarachnoid hemorrhage. In rare cases, where vasospasms occur after unruptured aneurysm surgery, the pathogenesis is most likely multifactorial and unclear. We present two cases of vasospasms following uncomplicated clipping of middle cerebral artery (MCA) aneurysms and a review of literature. Early diagnosis and therapy of this rare complication are necessary to achieve optimal clinical outcomes.

## 1. Introduction

Cerebral vasospasm (CVS) frequently complicates the course of patients with subarachnoid hemorrhage (SAH) caused by ruptured intracranial aneurysms. Delayed ischemic neurological deficit (DIND) accounts for 10–20% of mortality and morbidity in these cases [1,2]. Vasospasms after acute SAH typically begin in the first week after aneurysmatic rupture, and their severity decreases during the second week. The exact pathophysiology of cerebral vasospasms, despite intensive research over the last fifty decades, is still unknown. The generally accepted consensus focuses on the major role of blood breakdown products. This theory is supported by the relationship between the amount of the subarachnoid blood and the incidence and severity of vasospasms [3,4]. Delayed ischemic neurological deficits as a result of vasospasms after uncomplicated surgery of unruptured intracranial aneurysms (UIA) have rarely been reported (Table 1). The aim of the study was to present two cases, along with a complete review of identical cases without evidence of SAH (preoperative computed tomography (CT) scans, cerebrospinal fluid (CSF) evaluation, intraoperative findings), absence of intraoperative events (intraoperative rupture, twisting or destruction of parent vessels, diffuse bleeding), and negative postoperative CT images.

## 2. Case Report 1

A 65-year old woman, non-smoker, with history of arterial hypertonic disease and cardiac atrial fibrillation, was evaluated by computed tomography angiography (CTA) due to suspected internal carotid stenosis. There was no preceding headache or symptoms associated with SAH in the patient history. CTA excluded internal carotid artery (ICA) stenosis, but showed a left middle cerebral artery (MCA) aneurysm in the M1–2 bifurcation with a diameter of approximately 10 mm. The patient underwent uncomplicated surgery via a left lateral supraorbital craniotomy. There was no evidence of previous SAH during aneurysm dissection. The aneurysm was occluded with two curved titanium clips without intraoperative rupture. Intraoperative indocyanine green angiography (ICG) and direct micro-Doppler evaluation were performed to exclude stenosis of both M2 segments. The aneurysm sac was perforated after verified exclusion, and a very small amount of blood was immediately aspirated (Figure 1).

The application of temporary clips was not necessary. The postoperative course was uneventful. Postoperative CT and CTA performed 1 day after the surgical procedure were without evidence of hemorrhage or other complications. The aneurysmatic sac was completely excluded, without any residuum or signs of parenteral vessel stenosis (Figure 2).

On the 5th day, during the discharge process, the patient developed aphasia and right sided motor epileptic seizures. An acutely performed CT excluded any major complications; however, CTA revealed vasospasms in both M2 segments of the left middle cerebral artery. Mean velocity measured by Transcranial Doppler (TCD) was 150 cm/s on the left M1 segment. The patient was transferred to the angiography unit, and chemical intra-arterial angioplasty was performed with nimodipine according to our previously published protocol [5]. The diameter of both M2 segments, as well as the clinical status of the patient improved immediately during the procedure. Mean velocity decreased to 125 cm/s. Nimodipine was then administered intravenously (1 mg/h), and mean arterial pressure was maintained above 90 mmHg. Chemical angioplasty was repeated on the third day, due to another episode of clinical deterioration and increased mean velocity of 170 cm/s (Figure 3).

Milrinone was used for the second angioplasty and continued intravenously for six days, after which the medication was switched to an oral form of nimodipine. The patient’s clinical status improved throughout the two days after the second angioplasty. MR evaluation with diffusion-weighted images did not reveal any acute ischemic lesions. The patient was discharged home 16 days after clinical manifestation of vasospasm. She was without any neurological deterioration or other pathologies on follow-up CT and CTA one year after the procedure (Figure 4).

## 3. Case Report 2

A 72-year old man, non-smoker, with a history of arterial hypertonic disease, depression, and vertigo, underwent a diagnostic CT and CTA. There was no preceding headache or symptoms associated with SAH in the patient’s history. CT angiography showed bilateral MCA aneurysms in the M1–2 bifurcation with a diameter of approximately 8 mm on the right side and 7 mm on the left. The patient underwent uncomplicated surgery via a lateral supraorbital craniotomy on the left side after previous uncomplicated clipping of the contralateral right MCA aneurysm three months earlier (Figure 5).

There was no evidence of previous SAH during both aneurysm dissections. The aneurysm was occluded with two straight titanium clips without intraoperative rupture. An intraoperative ICG and direct micro-Doppler evaluation were performed to verify occlusion and exclude stenosis of both M2 segments. The aneurysm sac was perforated, and a very small amount of blood was immediately aspirated. Temporary clip application was not necessary, similarly to the first case. The postoperative course was uneventful. Routine postoperative CT and CTA performed one day after the surgical procedure was without any evidence of complication or hemorrhage. The aneurysmatic sac was completely excluded, without any residuum or signs of parenteral vessel stenosis (Figure 6).

The patient was discharged on the 5th postoperative day. During the next day, the patient developed global aphasia and disorientation. The patient was transferred to a local cerebrovascular unit to rule out stroke. An acutely performed CT excluded any complications; however, CTA revealed vasospasms in the M2 segments of the middle cerebral artery on the left side. The patient was transferred back to our center one day later. Mean velocity measured by TCD was 140 cm/s on the left M1 segment. The patient was transferred to the angiography unit and chemical intra-arterial angioplasty was performed with milrinone. Angiography confirmed peripheral vasospasms located in M4 segments (Figure 7).

The patient’s clinical status improved immediately during chemical angioplasty, and mean velocity decreased to 120 cm/s. The patient had to be intubated for recurrence of prolonged epileptic seizures. Milrinone was then administered intravenously (1 µg/kg/min) for six days, and mean arterial pressure was maintained above 90 mmHg. On the 6th day, milrinone was switched to an oral form of nimodipine. The patient’s clinical status completely improved throughout the seven days after angioplasty. MR evaluation with diffusion-weighted sequences revealed a small acute ischemic lesion in the ipsilateral MCA territory. The patient was discharged home on the 16th day after clinical manifestation of vasospasm without any neurological deficits. At the one year follow-up, the patient merely complained of fatigue and depression. Follow-up CT and CTA examinations did not reveal any pathology apart from the previously known small ischemic lesion in the semiovale center (Figure 8).

## 4. Discussion

Although common after acute subarachnoid hemorrhage, CVS and DIND are extremely rare following surgical clipping of UIA. During the pre-CT era, there were several published cases of potential DIND after UIA clipping [2,6,7,8,9]. These first published cases typically lacked preoperative CT or MR evaluation, as well as spectrophotometric analysis of cerebral spinal fluid (CSF). Preoperatively, many of these patients complained of headache or a newly present neurological deficit. Furthermore, there was a high incidence of intraoperative rupture and bleeding in these cases. Postoperative vasospasms typically occurred several hours after the initial procedure. Peerless in 1980 evaluated 53 cases DIND in cases of UIA published in literature; however, only eight cases were most likely true expressions of non-hemorrhagic vasospasm [7]. Fein evaluated 14 analogous cases during the same period, and explained the presence of CVS by intraoperative bleeding of a previously UIA [6]. Raynor presented the case of a young 25-year-old woman with suspected preoperative aneurysm rupture due to history of recurrent severe headache, photophobia, nerve palsy, and signs of CVS on the first angiographic evaluation before surgery [8]. Friedman presented a similar case in 1983. A 30-year-old woman developed complete 3rd cranial nerve palsy along with nausea, vomiting, and headache several days prior to surgery of a posterior communicating artery (PcomA) aneurysm [10]. Her clinical status suggested preoperative SAH despite a negative CT and lumbar puncture with “clear and colorless” CSF.

Bloomfield and Sonntag presented most likely the first true DIND after uncomplicated surgery of an unruptured MCA aneurysm in 1985. The patient was a 54-year-old woman with no history suggestive of SAH and had a negative preoperative CT. The patient developed left side hemiparesis and seizures on the second day following discharge, nine days after surgery [11]. Since this first case of true delayed nonhemorrhagic vasospasm after uncomplicated clipping of UIA, we found 14 more published cases [12,13,14,15,16,17,18,19,20,21]. All cases, including ours, are summarized in Table 1. 

Of the 17 published patients, only three reported thus far were men. Mean age was 57 years (range 21–72). All listed patients had DIND involving the anterior circulation. MCA aneurysms were most commonly affected in a total of 11 cases. Ophthalmic aneurysms, with four published cases, were in second place. The size of treated UIA ranged from 4 to 10 mm, with a mean sac size of 6 mm. The average onset of clinical deterioration due to CVS was eight days after uncomplicated clipping; however, the interval was very broad, ranging from 1 to 28 days. Vasospasm was diffusely located in two cases; in the remaining patients, it was localized near the aneurysm; and, in only one case, contralaterally [18]. A prolonged and unexpected headache after uncomplicated surgery of UIA may be a warning sign [16]. The treatment strategy depended on individual local CVS treatment guidelines. The partial or complete triple H therapy and chemical angioplasty were the most common therapeutic strategies used for DIND in UIA with subsequent oral administration of nimodipine [1,5]. Our treatment protocol for these rare cases is identical to CVS and DIND after SAH. Chemical intra-arterial angioplasty (nimodipine or milrinone) was performed as initial treatment with subsequent per oral administration of nimodipine based on our previously published protocol, as well as maintaining euvolemia and hypertension [5]. Regular monitoring, performed by TCD, and the evaluation of clinical status is absolutely mandatory. There is no data available concerning the possible benefit of local administration of vasodilator agents to prevent delayed CVS in cases of UIA surgeries. Half of the patients achieved full or good recovery; remaining patients had residual neurological deterioration (slight hemiparesis or aphasia). There were no published cases of patient death in this rare cohort. The prognosis was generally better than in SAH DIND cases, mostly likely due to absence of the bleeding insult. DIND and CVS after SAH is the leading cause of death of patients with aneurysm rupture after initial bleed, and occurs in 5 to 13.5% [22,23,24].

The mechanism of CVS and DIND has been studied for many years in the context of SAH. The degree of CVS was declared to be proportional to the volume of blood extravasated into the subarachnoid space [1,3,4]. Although no SAH was noted in this group of patients, a small amount of blood breakdown products released during elective uncomplicated surgery might be one of many factors involved in the pathogenesis of CVS. The blood breakdown products activate Rho kinase and protein kinase C pathways, both smooth muscle contraction agents [25,26]. Mechanical stress during preparation, clipping, and potential twisting of the clips, as well as the endothelium disfunction, are the second most discussed factors. Mechanical stress may play a role in evolution of intraoperative transitional CVS routinely presented during aneurysm surgery. The distribution of the CVS suggested that factors close to the UIA itself must play a role in the genesis of spasm. The endothelium stress may affect NO production and may release vasoactive agents such as endothelin-1, lipid peroxidase, and vasoactive cytokines [27,28,29]. The role of endothelium in production of prostacyclin, and potential overreaction mediated by vasoconstrictors prostaglandin, endoperoxide, and thromboxane A2, is again unclear [10]. The mechanical theory is supported by the paper of Schaller, who described the presence of vasospasms following Sylvian dissection [30]. On the other hand, our review showed that the application of multiple clips, such as temporary clips, which correlates with more extensive manipulation, was not a consistent risk factor. Temporary clipping was also not confirmed as a risk factor in the group of DIND after SAH [31]. Only Kitazawa demonstrated that number of clips and temporary occlusion of the ICA were found to be statistical determinants in his patient series [17]. Another historically described potential cause of CVS is the involvement of trigemino-cerebrovascular system (TCVS). Continuous stimulation by different stimuli (blood breakdown products, mechanical damage due to sac enlargement, preparation, temporary clips, etc.) may lead to a delayed depletion of vasodilatation substances after extensive activation [32,33,34]. Axons of this system reach the arteries via the ophthalmic and maxillary divisions of the trigeminal nerve. The TCVS system maintains normal vessel diameter via continuous depletion of substances such as substance P or calcitonin gene-related peptide (CGRP), which may be exhausted after prolonged stimuli [35,36,37]. Another theory explaining CVS is the occurrence of transient vasculitis due to an allergic reaction to metal clips (nickel, titanium). However, this theory was never confirmed by laboratory testing [16]. One of the oldest theories of CVS pathogenesis is the hypothalamic hypothesis, which suggests that mechanical or vascular compromise of the hypothalamus could promote the release of vasospastic mediators [38,39]. However, this theory does not explain CVS location, mostly in MCA aneurysms with a surgical corridor distal to midline structures.

## 5. Conclusions

DIND due to CVS is rarely encountered after elective uncomplicated surgery for UIA. The true incidence of CVS is most likely higher than reported in literature. The etiology of CVS in cases lacking subarachnoid hemorrhage is not clear, but most likely multifactorial. We described two cases of DIND after uncomplicated UIA clipping. CVS were successfully managed by chemical angioplasty in both cases. Early accurate diagnosis and active therapy were necessary to improve the probability of good outcomes in these patients. The prognosis was generally better than in SAH DIND cases, as documented cases do not describe any patient deaths or severe residual morbidity. Patients and clinicians should be informed about this rare condition to minimize treatment delay.

## Figures and Tables

**Figure 1 brainsci-10-00495-f001:**
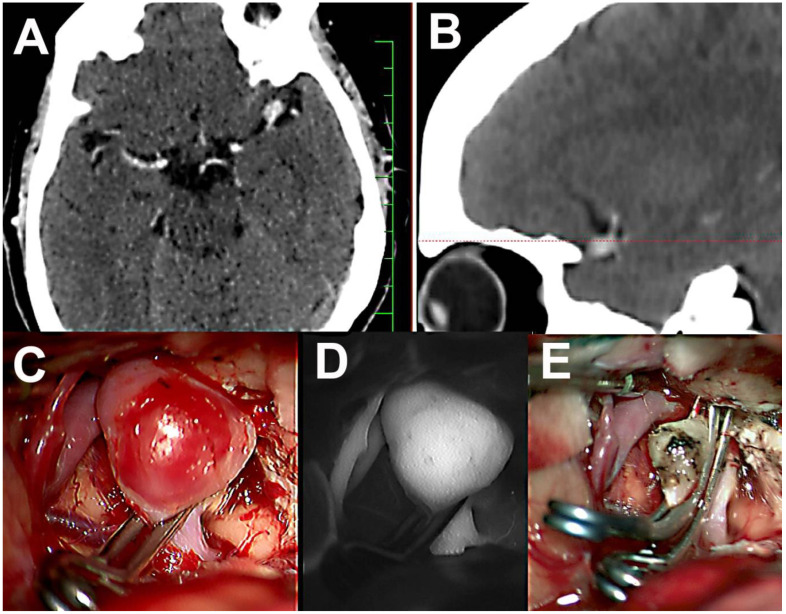
Preoperative computed tomography angiography (CTA) and perioperative microscope image together with indocyanine green angiography (ICG) evaluation following application of tandem curved clips. (**A**,**B**) Preoperative CTA of the left middle cerebral artery (MCA) bifurcation aneurysm. (**C**) Application of the first clip. (**D**) Persistent filling of aneurysm on ICG angiography. (**E**) Application of the second curved tandem clip for total closure and the resection of the sac.

**Figure 2 brainsci-10-00495-f002:**
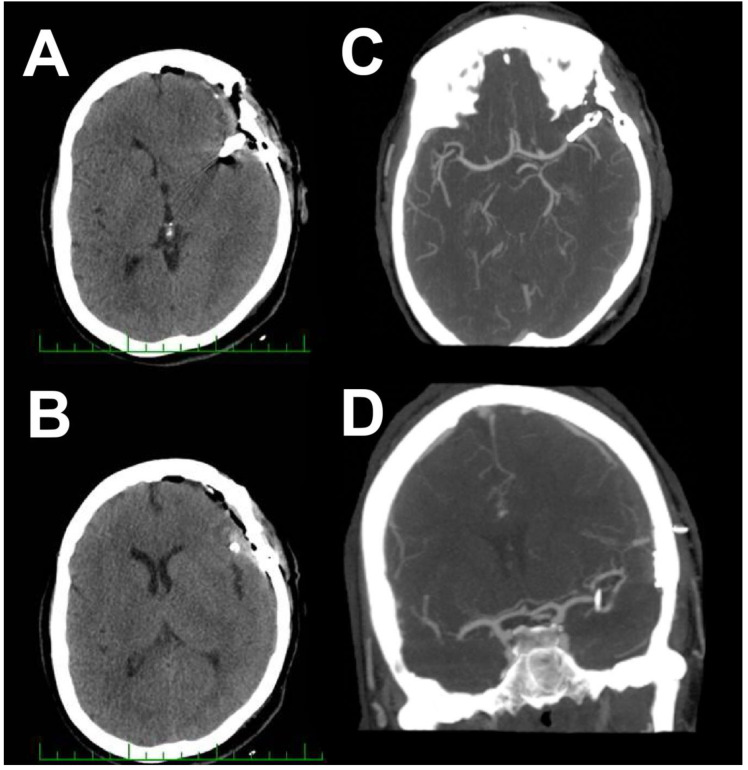
Postoperative CT and CTA evaluation one day after surgery. (**A**,**B**) Postoperative CT scan. (**C**,**D**) Postoperative CTA.

**Figure 3 brainsci-10-00495-f003:**
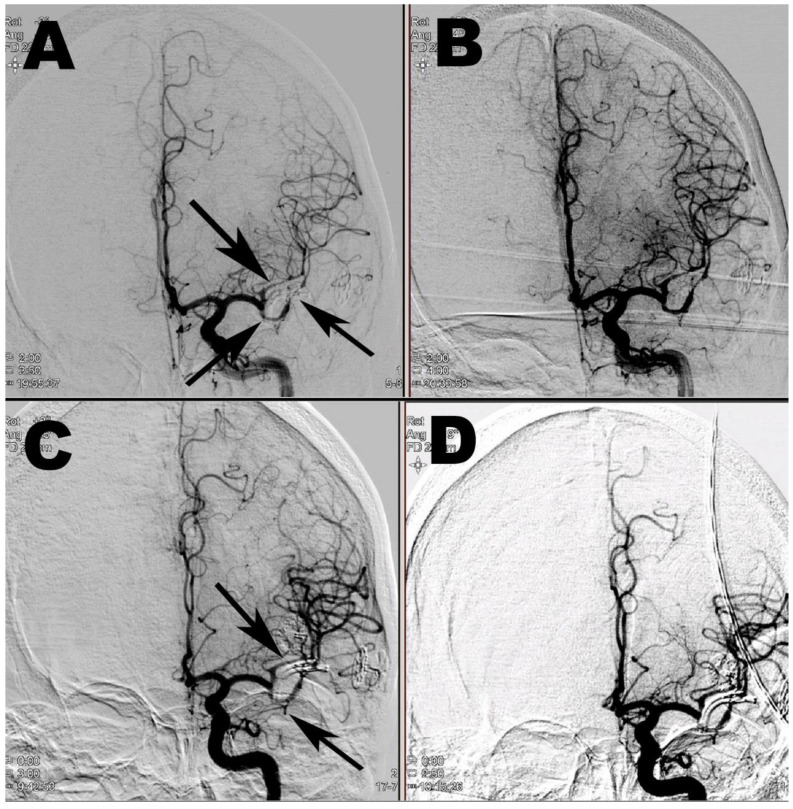
Chemical intra-arterial angioplasty. (**A**) DSA before the first chemical angioplasty with vasospasms in both M2 segments of the left MCA (arrows). (**B**) After intra-arterial nimodipine application. (**C**) Before the second chemical angioplasty with recurrence of vasospasms in both M2 segments of the left MCA (arrows). (**D**) After intra-arterial milrinone application.

**Figure 4 brainsci-10-00495-f004:**
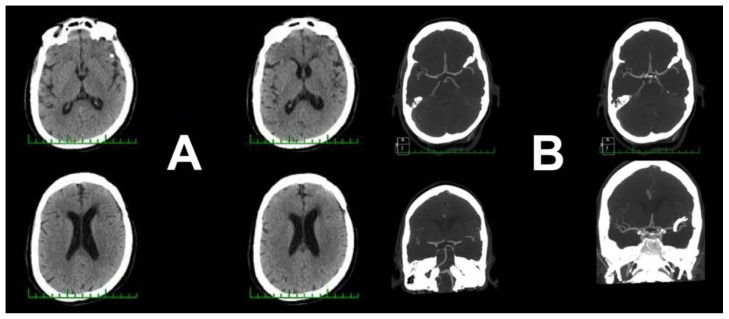
CT and CTA control one year after surgery. (**A**) CT scan and (**B**) CTA evaluation one year after the surgery.

**Figure 5 brainsci-10-00495-f005:**
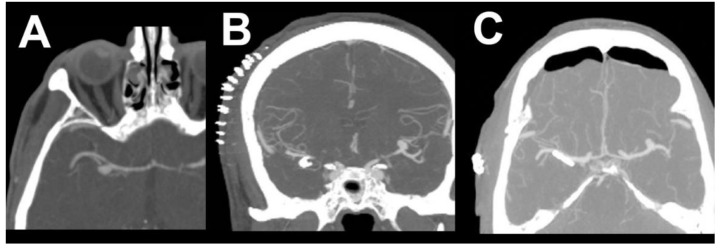
CTA of both MCA aneurysms. (**A**) Right MCA bifurcation unruptured intracranial aneurysm (UIA) before surgery. (**B**,**C**) CTA evaluation after uncomplicated right MCA UIA closure, left MCA bifurcation UIA is still obvious.

**Figure 6 brainsci-10-00495-f006:**
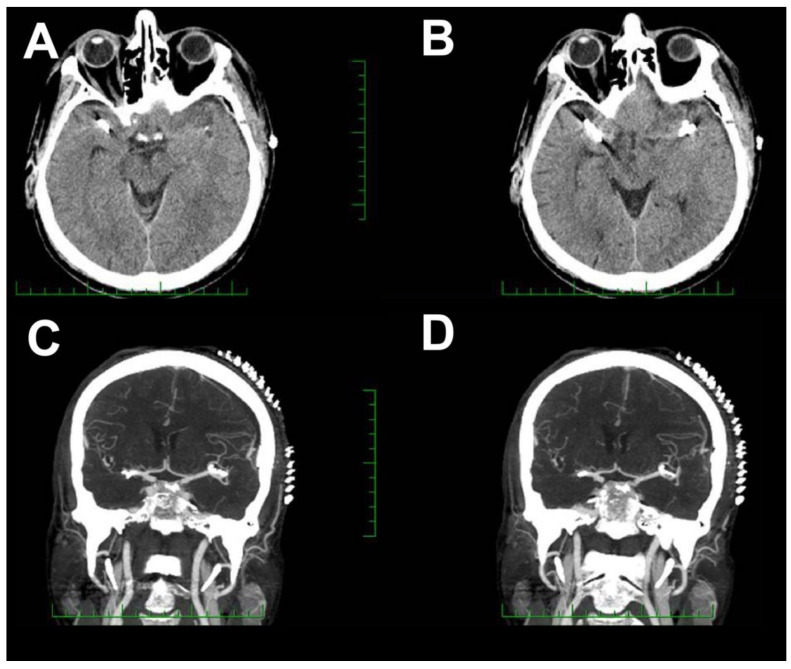
Postoperative CT and CTA one day after surgery. (**A**,**B**) CT scan one day after clipping of the left mirror aneurysm. (**C**,**D**) CTA without any signs of complication.

**Figure 7 brainsci-10-00495-f007:**
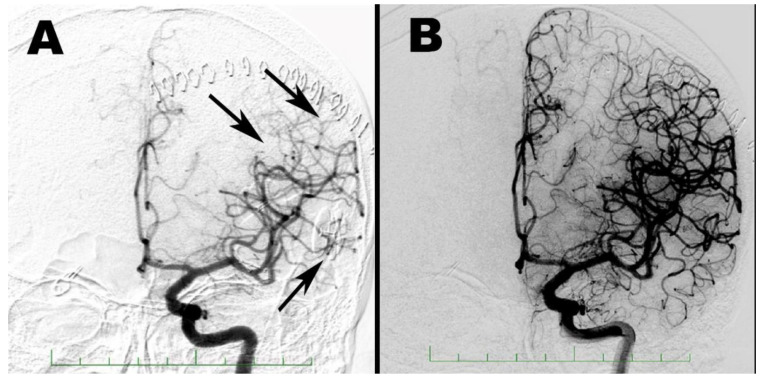
Angiography of the left carotid artery before (**A**) and after milrinone administration (**B**) with peripheral vasospasms located in M4 segments of the left MCA (arrows).

**Figure 8 brainsci-10-00495-f008:**
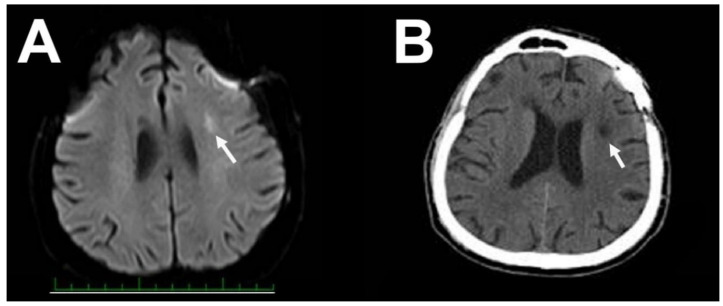
Small ischemic lesion in the semiovale center on acute diffusion-weighted MRI (**A**) and one year after surgery on CT scan (**B**).

**Table 1 brainsci-10-00495-t001:** Characteristics of published patients with uncomplicated surgery of UIA and subsequent cerebral vasospasm (CVS) and delayed ischemic neurological deficit (DIND) (ICA—internal carotid artery, MCA—middle cerebral artery, AcomA—anterior communicating artery, PcomA—posterior communicating artery).

Article	Age/Sex (year)	AN Location	AN Size (mm)	Clinical Presentation	Onset (day)	Temporary Clip	Intraoperative Event	Post-Operative CT Pathology	Treatment of CVS	Recovery
Bloomfield 1985 [11]	54/F	Right MCA	7	Hemiparesis	9	Unknown	Focal spasm	No	Hypervolaemia Steroids	Residual hemiparesis
Gutiérrez 2001 [15]	54/F	Left ophthalmic ICA	5	Aphasia Hemiparesis Coma	1	No	No	No	Papaverin	Residual hemiparesis
Kitazawa 2005 [17]	21/F	Left ophthalmic ICA	4	Aphasia, Gerstmann sy	12	Yes	No	No	Triple H Papaverin	Good recovery
Kitazawa 2005 [17]	63/F	Left ophthalmic ICA	5	Aphasia Hemiparesis	5	Yes	No	Mild EDH	Triple H	Good recovery
Kitazawa 2005 [17]	59/F	Left ophthalmic ICA	5	Aphasia Hemiparesis	2	No	No	No	Triple H	Good recovery
Paolini 2005 [19]	47/F	Right MCA	8	Hemiparesis	2	Yes	No	Small ICH	Hypervolemia Antiplatelet	Full recovery
Yang 2011 [21]	41/F	Right MCA	5	Aphasia Facial numbness	28	Yes	No	No	Hypervolaemia, antiplatelet, nicardipine	Residual aphasia
Yang 2012 [21]	61/F	Left MCA	6	Aphasia Mental change	10	Yes	No	No	Hypervolaemia, antiplatelet, nicardipine	Residual aphasia
Tsyben 2016 [20]	53/F	Left MCA	5	Aphasia Hemiparesis	2	No	No	No	Hypervolaemia, hypertension, verapamil	Residual hemiparesis
Tsyben 2016 [20]	70/M	Left MCA AComA	-	Unconsciousness Aphasia Hemiparesis	2	Only for ACoA	No	No	Hypervolaemia, hypertension, verapamil, nimodipine	Full recovery
Hashimoto 2016 [16]	62/F	Left PComA	5	Aphasia Disorientation Hemiplegia	11	No	No	No	Hypervolaemia, antiplatelet	Acalculia, paraphasia
Campe 2019 [12]	69/F	Right MCA	-	Hemiparesis Confused Motor speech disorder	12	No	No	No	Antiplatelet, nimodipine	Full recovery
Ceraudo 2020 [13]	59/F	Left MCA	5	Motor aphasia Hemiparesis	6	Yes	No	No	Nimodpine	Paraphasia, occasional speech arrest
Knight 2020 [18]	68/M	AComA	5	Facial drop Dysarthria Aphasia	5	.	No	No	Nicardipine, hypertension	Short-term memory deficit
Cuoco 2020 [14]	53/F	Left MCA	6	Hemiparesis Hemianopsia	13,26	Yes	No	No	Hypertension, verapamil, nimodipine	Hemiparesis, hemianopsia
Vachata 2020	65/F	Left MCA	10	Aphasia Hemiparesis Seizures	5	No	No	No	Hypertension nimodipine, milrinone	Full recovery
Vachata 2020	72/M	Left MCA	7	Aphasia Disorientation	6	No	No	No	Hypertension, milrinone, nimodipine	Fatigue, depression Deterioration

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
