# Peer review of "Delayed Ischemic Neurological Deficit after Uneventful Elective Clipping of Unruptured Intracranial Aneurysms"

_brainsci, 2020, doi:10.3390/brainsci10080495_

Round 1
Reviewer 1 Report
The authors present two cases of their own series that developed delayed ischemic neurological deficit due to vasospasm after clipping of unruptured aneurysms. Furthermore, a review of cases in the literature is followed by a discussion of the potential underlying pathophysiological aetiology of that phenomenon.
The topic is very interesting because cerebral vasospasm as a cause of ischemic stroke after uneventful elective aneurysm surgery seems to be a rare event that demands attention regarding appropriate treatment. The paper is well structured, it reads with a good flow and both cases are well illustrated. Nevertheless, I have several comments and questions:
1) Table 1 is concise and well organised. However, as it is supposed to include all studies on the topic until 2020, a quick search on pubmed reveals several papers missing, i.e.:
- Christin Campe et al. Vasospasm and delayed cerebral ischemia after uneventful clipping of an unruptured intracranial aneurysm - a case report. BMC Neurol 2019 Sep 16;19(1):226.
- Joshua A Cuoco et al: Recurrent Cerebral Vasospasm and Delayed Cerebral Ischemia Weeks Subsequent to Elective Clipping of an Unruptured Middle Cerebral Artery Aneurysm. World Neurosurg. 2020 May 31;141:52-58.
- James A Knight 2nd et al.: Contralateral Vasospasm in an Uncomplicated Elective Anterior Communicating Artery Aneurysm Clipping. World Neurosurg 2020 Jun;138:214-217
I would like to invite the authors to perform a more comprehensive literature research and include all the relevant case reports/series on the topic. In addition, formatting of the table may me optimised (no single letter on a new line)
2) Figure 3 and 7: An indicator (such as an arrow) would be helpful to show the spastic (or re-dilated) vessel position
3) It would be of much interest to elaborate deeper on the strategy behind the treatment protocol. Since the authors state that “the prognosis is generally better than in SAH DIND cases“ does their protocol differ in any way from treatment for those patients with vasospasms after SAH?
4) The authors’ treatment protocol is described as the application of milrinone (intravasally) followed by nifedipine (per os) in case of spasm. Have there been any cases described developing spasms after intraoperative application of nimodipine? Since excessive manipulation of the vessel is hypothesized to be a potential underlying factor for the development of spasms, is a preventive therapy (i.e. intraoperative nimodipine) justified in these cases in the authors eyes?
5) There are occasional inaccuracies in English grammar, particularly with (missing) preposition and erroneous wording. Please proof read. (I.e line 86/131: I assume the authors refer to “diffusion-weighted” imaging, rather than “diffuse weighted” and “sequences” instead of “sentences”?)
Author Response
Thank You very much for your valuable comments! All recommendations were accepted.
- Complete update of review was performed by both universities. The previous was done at the end of 2019. New articles are included in the manuscript.
- Both figures include arrows.
- The treatment protocol was commented.
- The sentence about intra-operative local application of vasodilatatory agents is included in discussion.
- The errors mentioned by reviewer were corrected. The whole manuscript was again carefully reviewed by coauthor Dr. Lodin (native English speaker from British Columbia).
Reviewer 2 Report
First, I appreciate the opportunity to review this interesting case series report.
The cases were well-described in general with sufficient background research.
I think this report is qualified enough to be published after some minor revisions.
Below are my minor recommendation for your report.
- Aim of study in introduction section (line 39-45)
- Currently, clear aim of study (we report~) was shown and then the experience of the authors was described. I think it will be better if you modified this into some compact and clear sentences without description of the author's experience.
- Use of non-standard abbreviation without prior suggestion of full-term
- line 155: Pcom
- Figure legends: Generally, every figure legend should be understrandable independently. Hence, abbreviations in every figure legends such as CT, CTA, and MCA need to be defined.
Author Response
Thank you for your valuable comments. All recommendations were accepted.
- Aim of study was corrected.
- All abbreviations were corrected and prior suggestion was done.
Round 2
Reviewer 1 Report
All my concerns form the previous version very satisfactorily revised. Thank you in particular for updating the table.